# Histone Sample Preparation for Bottom-Up Mass Spectrometry: A Roadmap to Informed Decisions

**DOI:** 10.3390/proteomes9020017

**Published:** 2021-04-21

**Authors:** Simon Daled, Sander Willems, Bart Van Puyvelde, Laura Corveleyn, Sigrid Verhelst, Laura De Clerck, Dieter Deforce, Maarten Dhaenens

**Affiliations:** 1Laboratory of Pharmaceutical Biotechnology/ProGenTomics, Ghent University, Ottergemsesteenweg 460, 9000 Ghent, Belgium; simon.daled@ugent.be (S.D.); Bart.VanPuyvelde@UGent.be (B.V.P.); Laura.Corveleyn@UGent.be (L.C.); Sigrid.Verhelst@UGent.be (S.V.); Laura.DeClerck@UGent.be (L.D.C.); dieter.deforce@ugent.be (D.D.); 2Department of Proteomics and Signal Transduction, Max Planck Institute of Biochemistry, 82152 Martinsried, Germany; willems@biochem.mpg.de

**Keywords:** histone code, coverage, epigenetics, mass spectrometry, sample preparation, workflow optimization, GingisREX

## Abstract

Histone-based chromatin organization enabled eukaryotic genome complexity. This epigenetic control mechanism allowed for the differentiation of stable gene-expression and thus the very existence of multicellular organisms. This existential role in biology makes histones one of the most complexly modified molecules in the biotic world, which makes these key regulators notoriously hard to analyze. We here provide a roadmap to enable fast and informed selection of a bottom-up mass spectrometry sample preparation protocol that matches a specific research question. We therefore propose a two-step assessment procedure: (i) visualization of the coverage that is attained for a given workflow and (ii) direct alignment between runs to assess potential pitfalls at the ion level. To illustrate the applicability, we compare four different sample preparation protocols while adding a new enzyme to the toolbox, i.e., RgpB (GingisREX^®^, Genovis, Lund, Sweden), an endoproteinase that selectively and efficiently cleaves at the c-terminal end of arginine residues. Raw data are available via ProteomeXchange with identifier PXD024423.

## 1. Introduction

The eukaryotic lineage arose from the symbiotic merger between bacterial and archaeal cells. Herein, the bacterial partner contributed genes, metabolic energy, and the building blocks of the endomembrane system, while the archaeal partner provided the potential for complex information processing by adding histones to the eukaryotic experiment [1]. The strongly conserved histone proteins provided the potential for complex information processing by developing a nucleosome-based chromatin structure [2,3]. This, together with the complex language of chemical modifications, evolved into the so-called histone code [4]. This not only enabled the organization of vast amounts of DNA, but also the regulation of gene expression through the modulation of the functional state of DNA [1]. In fact, it is hypothesized that the modifications of the histone code arose as a sensing mechanism of the energetic state of the cell; the presence of energy-rich donors like acetyl-CoA (acetylation), ATP (phosphorylation), and SAM (methylation) enabled direct chemical modification of the histone backbone [5], thus fully entangling the Prokaryote energy housekeeping with the Archaeal expression system. Importantly, these epigenetic changes can persist over many cell generations, allowing for the differentiation of stable gene-expression patterns in various cell types, essential for multicellular organisms [6]. In other words, histones are at the heart of the eukaryotic information management system and therefore amongst the most complexly modified proteins in the biotic world.

Over the past decades, it has become clear that mass spectrometry (MS) is an indispensable technique capable of studying the complex interplay of the histone code [7], i.e., the proverbial grammar of the Eukaryote information management system. In essence, an LC–MS/MS system measures the intensity and certain physicochemical properties of analytes, like mass-to-charge (*m*/*z*), retention time (RT), drift time (DT) and fragmentation pattern. This allows the creation of a multidimensional data space that captures a picture of the full complexity of the PTM landscape purely built from numbers.

However—as for any proteomics workflow—all parts of a histone workflow are entangled, and each combination of sample preparation, data acquisition and data analysis provides a different image of the histone code. Top-down and middle-down proteomics can omit or simplify many steps in sample preparation workflows, but the combinatorial complexity of coexisting PTMs makes many proteoforms hard to resolve [8,9,10]. Although middle-down approaches are more feasible and are particularly gaining interest in the field of LC–MS/MS-based histone analysis [11,12], bottom-up approaches remain most widely used.

Unfortunately, the gold standard for LC–MS/MS-based proteomics, i.e., tryptic digest, is less well suited for a large part of the histone code, due to the high abundance of lysine (K) and arginine (R) cleavage sites. Unmodified K and R residues result in peptides too small for efficient LC-separation [8,13], while changes in the modification status of K (and to a lesser extent R) dynamically block digestion, which in turn creates peptidoforms of different lengths complicating the alignment between runs for label-free quantification. Taken from the general abundance of both cleavage sites and their modification frequencies, a digest with arginine specificity would reduce these problems considerably. To this end, two widespread workarounds have been described. On the one hand, there is enzymatic digestion with clostripain, also known as ArgC—an enzyme that naturally cleaves at the c-terminal end of arginine residues. However, its lack of specificity discredited it for most users [14]. On the other hand, lysine residues can be chemically derivatized to obstruct digestion with trypsin, thus incorporating its superior efficiency and specificity into the workflow while obtaining an arginine-specific cleavage [15]. This most commonly used strategy gives good coverage of the N-tail of histone H3 and H4, but lags behind on coverage of other regions in the histone code and introduces additional chemical noise into the workflow [16,17,18,19,20]. Overall, the histone coverage greatly depends on the sample preparation workflow, regardless of how advanced the LC–MS/MS system and posterior data analysis might be. Indeed, it is currently impossible to cover the entire histone code with a single workflow.

Here we present a new perspective on histone sample preparation that guides researchers in selecting a protocol that covers a specific stretch of interest while taking certain limitations such as side reactions into account. As a proof-of-concept, we have acquired data from bovine histone standards prepared with four different histone sample preparation workflows, including a workflow with the arginine-specific enzyme RgpB (GingisREX, GRX, Genovis, Lund, Sweden), which has not been described for bottom-up LC–MS based shotgun proteomics applications before [21,22]. Additionally, digestion with trypsin, both with and without derivatization of the lysine cleavage sites, as well as digestion with ArgC, were included in the experimental design. To allow for a direct comparison of the obtained coverage, all treatments were executed in five replicates and measured in one randomized sample list with pairwise mixtures interspersed as quality control samples (QCs). An in-house tool was built that creates a snapshot of the histone coverage—including post-translational modifications (PTMs)—of both in-house and public LC–MS-based histone workflows. Additionally, the tool can be used to visualize the coverage of predicted MS/MS spectra, which enabled us to create coverage plots for nine histone PTMs using their curated depiction from Uniprot. This comparison between the curated histone code from Uniprot and the experimental histone workflows reveals the discrepancy between what is possible and what is feasible with the current-day workflows. Indeed, the experiments show that the coverage of all workflows is incomplete but complementary and that optimization of histone sample preparation is still very much alive and relevant. We therefore provide a stepping stone to an informed decision-making strategy regarding sample preparation and experimental design for bottom-up histone mass spectrometry workflows.

## 2. Materials and Methods

### 2.1. Sample Preparation

A bovine histone standard extracted from calf thymus (Sigma-Aldrich, Darmstadt, Germany, 10223565001) was used to avoid bias towards specific regions of the histone code caused by the extraction of histone proteins from cell lines or tissue. All protocols were executed five-fold on 20 µg of histone standard.

The ArgC and RgpB protocols—hereafter embedded in and referred to as the ArgC and GRX workflow respectively—were executed according to the instructions of the manufacturer. In brief, 5 µg ArgC (Sigma-Aldrich, Darmstadt, Germany, 11370529001) was resuspended in 50 µL ultrapure water (18.2 MΩ.cm resistivity), of which 4 µL was added to the samples, followed by 10 µL of activation solution (50 mM Dithiothreitol, 5 mM EDTA) and addition of digestion buffer (100 mM TrisHCL, 10 mM CaCl_2_) to a final volume of 100 µL. For the RgpB protocol, cysteine and dithiothreitol were added to the samples to a final concentration of 20 mM and 10 mM, respectively. The samples were incubated at room temperature for 30 min. Next, 5 µg RgpB (Genovis, GingisREX^®^, Lund, Sweden) was resuspended in 50 µL ultrapure water of which 4 µL was added. All samples were incubated overnight at 37 °C and vacuum-dried.

The tryptic digest samples—hereafter embedded in and referred to as the NoPropTryp workflow—were resuspended in 500 mM Triethyl Ammonium Bicarbonate buffer with an aimed final volume of 50 µL, followed by the addition of CaCl_2_ and acetonitrile to a final concentration of 1 mM and 5% (*v*/*v*), respectively. Finally, trypsin was added in a 1:20 (*w*/*w*) ratio and the samples were incubated at 37 °C overnight and vacuum-dried.

The tryptic digest with propionylation of the lysine residues—hereafter embedded in and referred to as the PropTryp workflow—has been described previously by Meert et al. [16]. In brief, the samples were resuspended in 20 µL TEAB. First, 20 µL of an isopropyl alcohol: propionic anhydride (7:1) solution was added, followed by incubation at room temperature for 30 min. Next, 20 µL of ultrapure water was added, followed by incubation at 37 °C for 30 min to quench the reaction. The samples were vacuum-dried and digested as described for the tryptic digest. After digestion, the N-termini of the resulting peptides were propionylated accordingly and the samples were vacuum dried. To reverse over-propionylation on serine, threonine, and tyrosine residues, 50 µL 0.5 M hydroxylamine and 15 µL ammonium hydroxide (pH 12) were added, followed by incubation for 20 min at room temperature. The samples were acidified (pH 3) with formic acid and vacuum dried.

### 2.2. Sample Acquisition

All samples in the sample list were randomized to avoid systematic variation, blocked to correct for drift in sensitivity of the instrument, and interspersed with quality control samples to monitor instrumental variation. Seven different quality control samples were prepared, four of which were mixtures of all separate digest conditions. In addition, a mixture of GRX and ArgC samples; a mixture of NoPropTryp, ArgC, and GRX samples; and a mixture of PropTryp and NoPropTryp samples were prepared. These were used to increase alignment efficiency and quantitative accuracy during analysis with Progenesis QIP.

Three micrograms of each sample was LC-separated prior to MS analysis with a Nano Acquity system coupled to an Acquity UPLC CSH-C18 analytical column (1.7 µm, 300 µm × 100 mm, Waters) using a two-step linear gradient of 50 min (3–30% B) and 10 min (30–40% B), and 0.1% formic acid and 3% DMSO in water was used as solvent A and 0.1% formic acid in acetonitrile as solvent B.

All samples were acquired on a Synapt G2-Si (Waters) in HD-DDA mode. Herein, fragment ions are mobility-separated based on charge state, which allows the instrument to selectively sample single charged fragment ions in the ion beam that enters the TOF-tube. This significantly increases the duty cycle and thus sensitivity for these fragment ions. This is called wideband enhancement [23]. Precursor scan time was set to 0.2 s, and each scan cycle allowed for fragmentation of a maximum of 12 precursor ions with an intensity threshold of 3000, a scan time of 0.1–0.3 s and a switchback threshold of 100,000. The mass range was set to *m*/*z* 50–5000 for both precursor and fragment scans.

### 2.3. Coverage Plots

Spectral libraries were generated according to the workflow of Van Puyvelde et al. [24]. Briefly, a protein sequence FASTA file of all histones was in silico digested with trypsin specificity. Both double- and triple-charged peptide fragment spectra were predicted using MS^2^PIP [25], the results were written to a spectral library file (*.msp format). Retention time prediction using DeepLC was disabled, given the intended purpose of the library [26,27].

Using an in-house script (available at https://github.com/swillems/spectral_histones, accessed on 19 November 2020), all known and verified uniprot modifications (exported from uniprot as a .txt file) were parsed into the predicted spectral libraries and exported as an *.MGF file. Coverage plots can be created from both theoretical and empirical MGF files by searching the spectrum files in Mascot Daemon and exporting the results as a *.CSV file, including the start and end points of all identified peptides. The in-house coverage plot script (available at https://github.com/swillems/coverage_plots, accessed on 19 November 2020) also requires the protein database (*.FASTA) used to search the MGF files to map the identified peptides and PTM against the protein backbone. The 10 amino acid marker functionality was disabled in the figures produced in this manuscript. The displayed modifications depend on the search parameters used in Mascot Daemon.

### 2.4. Quantitative Analysis with Progenesis QIP

All samples were imported, peak picked, and aligned in Progenesis QIP (Nonlinear Dynamics) for quantitative comparison. Each experiment/experimental design had its accompanying QC samples for alignment and normalization purposes (Figure 1). For identification, all MSMS spectra were exported in a single *.mgf file for searching using Mascot (Matrix Science). A dedicated instrument was configured for the search, i.e., Synapt G2-Si, which only allows 1+ fragments because wideband enhancement was enabled during acquisition [23]. Bovine Histones and cRAP (available at https://www.thegpm.org/crap/, accessed on 9 December 2019) were downloaded from Uniprot on 9 December 2019 (141 sequences; 43,864 residues). Search parameters were selected based on the experimental design under investigation. The PropTryp-project was searched with Acetyl (K), Butyryl (K), Citrullination (R), Crotonyl (K), Dimethyl (KR), GG (K), Methyl (KR), Phospho (ST), and Trimethyl (K) as variable modifications and Propionyl (K) and (N-Term) as fixed. The ArgC-GRX-project and the GRX-project were searched with Acetyl (K), Butyryl (K), Citrullination (R), Crotonyl (K), Dimethyl (KR), GG (K), Methyl (KR), Phospho (ST), and Trimethyl (K) as variable modifications with two missed cleavages. After all identifications were imported back into Progenesis QIP, all LC–MS runs were normalized against histone peptides in order to consolidate the constant protein abundance within the data. Notably, for coefficient of variance calculations, no normalization was used as this would skew the results. Here, we relied on the fact that we started from an identical amount of histone sample at the beginning of each workflow.

### 2.5. Variant Calling

Mascot CSV-files were exported and only unique peptide spectral matches (PSMs) were retained to create a bar chart with all workflows and their respective unique PSMs per variant. These PSMs can only be attributed to one histone variant and are flagged by Mascot as unique.

### 2.6. Data Availability

The mass spectrometry proteomics data have been deposited to the ProteomeXchange Consortium via the PRIDE partner repository with the dataset identifier PXD024423 and 10.6019/PXD024423.

## 3. Results

Comparing the performance of sample preparation workflows requires a single experiment wherein samples are prepared and measured in a randomized way to avoid batch effects. This allows for an objective assessment of the histone coverage, the chemical noise and variation introduced by the workflow, the enzyme specificity and efficiency, the experimental variation, and the quantitative accuracy. Here, we assess these features for four different workflows (Figure 1), providing a roadmap for future workflow assessment in the process.

### 3.1. The Histone Code Coverage with Tryptic Specificity

#### 3.1.1. A Theoretical Tryptic Histone Snapshot

To enable direct comparison of the histone coverage of different sample preparation workflows, we first created a PTM Coverage Tool in Python. To illustrate its applicability, we created an overview of detectable curated modifications of the histone backbone with the search parameters depicted in Figure 2. This also assesses the efficiency of the search strategy itself. Therefore, a spectral library of unmodified histone peptides was predicted using MS^2^PIP [28], and all delta masses of curated histone modifications from Uniprot were parsed into the spectral library at their corresponding location. Note that these noiseless predicted spectra generate a considerable amount of ambiguity above the significance threshold when searched by Mascot using the set of variable PTMs shown. This is due to localization errors and unconsidered PTMs present in the MGF file that are explained through other PTM combinations or point mutations [29]. Therefore, all searches on empirically measured histone peptidoforms below should be considered in light of this first limitation. Despite this limitation, Figure 2 shows that a considerable part of the histone code can be covered theoretically using a workflow with tryptic cleavage.

Taken together, the tool presents the researcher with an end-point report of the full workflow, including the search. The yellow regions highlight where many different peptidoforms are created, which translates into higher spectral counts in a predicted spectral library as shown in Figure 2.

In search of a suitable protocol, the tool can also be used to visualize the detected PTMs in public data, with the current limitation that results need to be parsed in the Mascot *.csv format. As an example, Appendix A depicts the PTMs annotated using Progenesis QIP and Mascot on the AQUA heavy peptide standard as described in Lin et al. [30]. In doing so, the list of peptides in the AQUA mixture was converted to one image depicting the detected PTMs with the given search parameters.

#### 3.1.2. Theory Versus Practice

Figure 3 depicts the coverage plot of an empirical workflow using trypsin. Five independent tryptic digests of a commercial bovine histone extract were performed and analyzed using LC–MS/MS, interspersed with QC samples (Figure 1). This provides a very different picture compared to the predicted coverage plot. Indeed, only a fraction of the possible peptidoforms are identified.

This is primarily due to the fact that only a small subset of all possible proteoforms are actually present in the sample, i.e., a calf thymus histone extract. However, equally important is the fact that certain peptidoforms, which are present in the sample, are excluded from annotation because of a number of other reasons: (i) the enzyme could have created unconsidered aspecific cleavages, or these could have been present in the sample to begin with [31]; (ii) in vitro induced PTMs like deamidation, oxidation, and formylation were not considered in the search; (iii) some peptidoforms have no retention on the LC column or lie outside the mass range acquired by the instrument; (iv) singly charged precursor peptides were not targeted by the instrument in data-dependent acquisition (DDA) mode; (v) some peptidoforms are too short to generate enough specificity for the probabilistic scoring algorithm to reach the significance threshold; (vi) gas-phase chemistry artifacts (like neutral losses, cation adducts, and in-source decay) were not considered in the search.

Overall, it is clear that the epicenters of sequence coverage have shifted considerably, most strikingly so for H3. Still, the core and C-termini, including those of H2A, H2B, and H1 are sampled quite efficiently. This indicates that, although tryptic digest is usually not preferred for histone epigenetic research, it can be of value for certain regions in the histone code, especially when considering workload. Therefore, trypsin should always be the first choice if the coverage of the workflow suits the research question. Indeed, numerous histone-related discoveries have been described using this workflow. One such example was the identification of H2A-specific protease as being neutrophil elastase, capable of cleaving H2A at valine 114, i.e., five amino acids upstream the essential PRC1-mediated K119 ubiquitination [31,32,33].

#### 3.1.3. Exploring Other Options

This lack of H3 and H4 N-tail coverage in bottom-up histone research is most commonly addressed by blocking tryptic digest at the lysine cleavage site by derivatization with, e.g., propionic anhydride, thus introducing propionyl groups [15]. This results in larger and more hydrophobic peptidoforms, more suitable for LC–MS/MS analysis of those sequence stretches (Figure 4). Since these regions are densely covered with PTMs, this protocol has been the preferred approach for bottom-up histone analysis for nearly 15 years. However, the core and C-termini of most histones are less well/accurately sampled because of the oversized peptides. This not only impairs LC-separation for these peptides but also hampers in-depth fragmentation. Both increase ambiguity, which in turn hampers accurate identification and localization of PTM combinations with current search algorithms. Due to these erroneous annotations of isobaric combinations, two intensely sampled regions become overreported with PTMs: histone H2B 1-29 and H1 1–32 [29]. These suspicious cases can only be detected by manually verifying the sequence coverage (Appendix A). Overall, caution is required with longer peptide stretches, especially when combinatorial PTMs are considered.

Unfortunately, chemical derivatization also introduces chemical noise, which interferes with the quantification and identification of important PTMs [16,17,18,34]. Figure 4B shows the coverage plot of Histone H3 when searched with a selection of PTMs that includes in vitro introduced PTMs. Indeed, when propionylation is combined with trypsin, propionyl groups of biological origin can no longer be distinguished and mono-methylated lysines are propionylated (56.026 Da + 14.016 Da), which results in a mass shift identical to butyryl (70.042 Da), thus becoming indistinguishable.

The protocol can also introduce overpropionylation (S, T, Y) and formylation as a side reaction, apart from amidations not displayed here [16,17,18,19,20]. This compromises the correct biological interpretation of all these PTMs. While many solutions have been proposed [16,17,18,34], including more controlled reaction circumstances and the use of heavy labeled or non-biological reagents, entirely excluding side reactions will probably prove to be impossible. This was recently illustrated by the finding that formylation of serine and threonine can even be induced by 0.1% formic acid in the LC buffer [35]. Fortunately, these side reactions do not interfere with the relative abundance of single PTMs [18]. Still, they dilute the signal, creating uninformative precursors that take up acquisition time of an instrument that is acquiring in DDA [16] and aggravate the already complex issue of ambiguity in annotation [29].

### 3.2. Comparing Three Different Enzymatic Treatments with ArgC Specificity

#### 3.2.1. Side Reactions and Chemical Noise

Theoretically, ArgC (Clostripain) generates similar-sized peptides compared to PropTryp without the disadvantages of chemical derivatization. However, the lack of specificity of the enzyme has been known for years [14,36]. Therefore, we also validated another arginine-specific enzyme, i.e., RgpB, also known as Gingipain or GingisREX (Genovis, Lund, Sweden). This enzyme was first described in 1992 as an important virulence factor of Porphyromonas gingivalis and later praised for its cleavage specificity and efficiency at the c-terminal end of arginine [21,22,37]. However, it was only recently commercialized for proteomics applications. As expected, considerably less chemical noise was introduced in either of these workflows (Appendix A), while methylation, propionylation, and butyrylation remain useful for biological interpretation on all histones. Surprisingly, however, the sequence coverage of both enzymes shifts dramatically compared to PropTryp, as exemplified in Figure 5, which shows histone H3 for all the considered workflows as an example. Coverage plots of biological PTMs detected on other histones in each workflow are shown in Appendix A.

#### 3.2.2. Histone Variant Calling

Apart from histone PTMs, histone variants are also used by cells to regulate gene activity and chromatin structure. Intrinsically, the differences in backbone sequence make these variants interesting targets when considering sample preparation workflows. Figure 6 depicts the number of unique peptides for each of the variants identified in the commercial bovine histone standard. Most importantly, in the context of this study, are the backbone stretches that are covered by a certain sample preparation workflow, especially when these stretches are unique for a certain variant. It is clear that the ArgC workflow lags behind when it comes to variant calling. The other workflows require a more detailed view of the different variants. Therefore, Appendix A depicts the coverage plots of the histone H1 variants for Tryp, PropTryp, and GRX. This figure clearly shows that the Tryp-workflow has a better overall backbone coverage but lags behind on coverage of the theoretical modifications (Figure 2). PropTryp on the other hand has reasonable coverage of the N-terminal modifications but does not cover the rest of the backbone, which might hamper variant calling. Note that a combination of a histone PTM with an amino acid substitution can be isobaric and impossible to resolve if no (high-intensity) fragments are available to coin the correct annotation. Therefore, manual curation should be performed, but this lies outside the scope of this paper.

#### 3.2.3. Feature Detectability

In search of the characteristics that contribute to the shifting detectability, we moved beyond the PTM coverage tool. The experimental design (Figure 1) allowed for direct comparison of different treatments. Therefore, different Progenesis Projects were created. The PropTryp-project contained five PropTryp replicates and five quality-control (QC) injections. ArgC and GRX replicates were aligned directly in one project using a mixture of both treatments as alignment template and QC. A separate project was created for GRX that contained the GRX runs and their QC to study this enzyme in isolation. Alignment between runs at the MS1 level enables a more quantitative interpretation of the signal. Herein, all ions with a clear isotopic envelope and elution profile are defined as features to uncouple the quantitative interpretation from the acquisition and identification workflow used. This allows indirect analysis of the identification biases depicted under Section 3.1.2.

A total of 279 histone ions with the expected properties were annotated in the mixed ArgC-GRX project, i.e., specifically cleaved peptides at the c-terminus of arginine without missed cleavages. Forty-seven of these features were annotated through charge state deconvolution, a feature in Progenesis QIP that allows one to annotate singly charged features (signal with a clear isotopic distribution) that were not selected for fragmentation during DDA. To this end, the annotation of the doubly charged precursor is transferred by the software to the co-eluting, singly charged precursor-ion. For the PropTryp-project, 302 identifications were imported, of which 79 were found through charge state deconvolution.

Despite the presumed identical cleavage specificity, the three workflows show clear discrepancies. This is unsurprising for PropTryp because of the chemical derivatization. The two-dimensional LCMS representation of the three protocols (Appendix A), mainly shows the increased retention of the propionlyation reaction, whereby the different peptidoforms of the H3 and H4 N-tails are more efficiently separated and sampled, as described earlier [15].

However, GRX and ArgC also differ considerably. This is equally reflected in the principal component analysis of the feature abundances in the ArgC-GRX project (Figure 7A). An even more striking difference from the acquisition point of view is that the number of different MS1 precursors of all charge states is in the order PropTryp > ArgC > GRX, while the number of MS/MS-spectra is in the reversed order, GRX > ArgC > PropTryp (Figure 7B). The QC from the combined project illustrates that this is not an instrumental effect, as physically mixing ArgC and GRX results in intermediate values. At least in part, this turns out to be a consequence of the charge state distribution of the precursors. Figure 7C shows that nearly 50% of the PropTryp ions is singly charged as opposed to only 20% for the other two enzymes. This is likely caused by the charge blocking effect of the propionyl group and will exclude these precursor ions from fragmentation during regular DDA acquisition.

Indeed, the reduced sampling is also reflected in the summed MS1 signal of all identified peptidoforms (Figure 7D). Appendix A displays the charge state distribution of these annotated ions, confirming the overall pattern of all features depicted in Figure 7C. For GRX, the singly charged features annotated by deconvolution represent only 10%, as opposed to 40% for PropTryp. At the precursor ion level, the implications are further highlighted. Figure 7E illustrates how, e.g., the H3K9-R17 peptide stretch is about 60% singly charged on a SynaptG2Si micro flow source with 3% DMSO in the aqueous buffer. This implies that 2/3 of the signal cannot be annotated and is usually not used for quantification for this peptide. While, in time, data-independent acquisition (DIA) like the recent hSWATH workflow could recover this signal, targeting singly charged precursors in DDA on pure histone extracts could allow for a more conventional solution [38].

In conclusion, the differences in retention and charge state distribution together make GRX and PropTryp cover complementary sequence stretches, as visualized in Figure 7F.

#### 3.2.4. Enzyme Specificity and Efficiency

On top of the increased retention of the histone peptides, the propionylation protocol is also preferred because it uses trypsin as a proteolytic enzyme, which assures high efficiency and specificity. To assess the enzyme specificity of ArgC and GRX, the same data were searched with semi-specific cleavage. This results in a total of 1135 different annotated histone features, 142 of which by charge state deconvolution. Appendix A shows that this indeed provides a better histone coverage for ArgC.

However, there is an important caveat: these semi-specific peptides cannot readily be used for biological interpretation, as adding them to the dataset would significantly alter normalization, which would in turn impact the relative abundance calculations of individual PTMs. Indeed, when based on all detected features, the calculated log normalization factors between the enzymes are close to zero for all runs. This is an intuitive consequence of the fact that all workflows were executed on an identical amount of commercial bovine histones (Figure 8A). However, Figure 7D already showed that for the identified desired features with specific cleavage, only half the signal was found in ArgC compared to GRX. This implies that using only the correctly cleaved ions to normalize the data results in large normalization differences between ArgC and GRX runs (Figure 8A). Normalizing against semi-ArgC-specific annotated peptides reduces this gap but does not remove it completely. Thus, ArgC proportionally has more identified aspecifically cleaved peptides, while the rest of the gap can be explained by further degradation of the sample into peptides that can no longer be annotated by the applied searches. To assess the enzyme efficiency of ArgC and GRX, the data were searched with up to two missed cleavages. In total, 83 (7%) additional features were identified, most of which more abundant in ArgC, implying that this enzyme is also less efficient (Figure 8B). However, while only 8% (98/1246) of all annotated ions had a citrullination on arginine, this fraction became 35% (29/83) for the peptides that contained missed cleavages. This also holds for the 3.6% (45/1246) arginine methylations in the total population that is enriched to 23% (19/83) in the missed cleaved population. This reveals an alternative conclusion, i.e., that in fact GRX is capable of cleaving modified arginines and ArgC is not.

Taken together, because of its comparable efficiency and higher specificity, GRX generates larger peptides compared to ArgC (Figure 8C). However, more surprisingly, the detected peptides in GRX are also considerably longer than what is detected with PropTryp. This is mainly due to the increased hydrophobicity induced by propionylation. More specifically, where this increased hydrophobicity is a benefit for the retention and coverage of hydrophilic short and mid-range peptidoforms of H3 and H4 N-tails, it requires alternative LC strategies to resolve the longer peptides—again at the cost of the shorter ones. The result is that GRX covers overall a larger size range of histone peptides but at the cost of resolution or even retention of the densely modified H3 and H4 N-tail peptidoforms. These larger peptides urge for alternative fragmentation and annotation strategies for the GRX workflow that are akin to middle-down approaches, while the short hydrophilic peptides require derivatization to increase hydrophobicity. Indeed, longer peptide stretches as depicted in Appendix A tend to only fray on the ends and do not fragment towards the middle when applying collision-induced dissociation (CID). We are therefore developing an adjusted GRX workflow, including phenyl iso-cyanate derivatization and electron transfer dissociation (ETD), which should respectively increase retention of smaller peptides and fragmentation of larger peptides.

As a concluding overview, Figure 9 shows the LC–MS feature map of the PropTryp and GRX enzymatic treatments with the peptide charge states color-coded. Appendix A highlight the localizations of the most prominent sequence peptidoforms identified in PropTryp and GRX, respectively, providing a true roadmap to these two enzymatic treatments.

### 3.3. Workflow Variability

Workflow variability—expressed as % covariance (%CV)—is one of the most important metrics to assess during validation of any sample preparation workflow, as it greatly impacts quantification and the required number of replicates in the experimental design [39]. Minimizing this variability becomes even more important when it comes to detecting small biological changes, as is often the case for histone PTMs. It is assumed that longer workflows usually introduce more variation. However, Figure 10A shows that the total %CV for each workflow is comparable, despite the additional derivatization steps in the PropTryp workflow. We therefore isolated the instrumental %CV by calculating the %CV on all precursor ions in the QC samples of each treatment (Figure 10B). Strikingly, while instrumental variation is generally assumed to be constant, the instrumental %CV of ArgC and especially GRX was higher compared to the NoPropTryp and PropTryp workflow. To avoid bias, the total %CV (light-colored graph) and instrumental %CV (dark-colored graph) as a function of precursor ion abundance were plotted (Figure 10C), which shows that instrumental variation is higher throughout the dynamic range for ArgC and GRX. This leads to the finding that the different workflows generate a comparable total %CV, but derived from different sources, i.e., sample preparation and instrumental %CV. We hypothesize that the larger peptidoforms from the GRX workflow are prone to a higher instrumental %CV because of their broad and possibly unstable charge state distribution. In that case, blocking of these higher charges by propionylation of K residues can reduce this effect.

## 4. Conclusions

Both trypsin and GRX have high specificity and efficiency, which makes them suited for histone and proteomics workflows in general. This is in contrast to ArgC, whose lack of specificity hampers the accurate quantification of peptides and peptidoforms. Without derivatization, trypsinization can be used to study the core and C-termini of all histones. Still, propionylation remains the preferred protocol to target the N-termini of histone H3 and H4, despite the chemical noise that is introduced by derivatization and the accompanying loss of several biological interpretations. We here also describe a strong shift in the charge-state distribution following propionylation, excluding many precusors from sampling in a DDA MS run. Surprisingly, the latter also reduces the instrumental variation in the workflow, which is remarkable since this source of variation is generally assumed to be constant. GRX on the other hand can be considered as a complementary enzyme with the specific benefit of retaining all biological hPTMs and minimizing chemical noise. It is particularly suited for detecting longer peptide stretches, which in turn might require more dedicated fragmentation and annotation strategies. The lack of retention of H3 and H4 N-tail peptidoforms reduces the sampling efficiency of these biologically essential sequence stretch. However, overall, short peptides are currently lost in GRX, while long peptides are lost in PropTryp.

## Figures and Tables

**Figure 1 proteomes-09-00017-f001:**
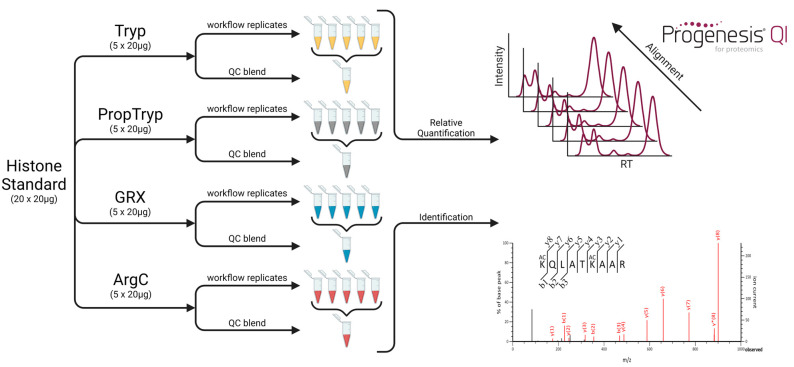
Experimental design for direct comparison of different enzymatic treatments. All replicates (*n* = 5) were prepared in parallel, and the samples were analyzed in a full factorial design sample list to avoid batch effects. The different QC mixtures were used for direct alignment between runs in Progenesis QI for Proteomics, which allowed for relative quantification. Mascot was used for the identification of which the exported CSV-files were used to create the coverage plots presented below. Next to the indicated QC samples, GRX-ArgC, Tryp-PropTryp and Tryp-GRX-ArgC blends were also prepared. These allowed for alignment between different workflows. Tryp: Workflow based on regular tryptic digest, PropTryp: Workflow based on the blocking of lysine residues with propionyl groups to obtain arginine specific cleavage during tryptic digest, GRX: Workflow based on digestion with RgpB (Genovis, GingisREX, Lund, Sweden), ArgC: Workflow based on digestion with clostripain (Roche, ArgC).

**Figure 2 proteomes-09-00017-f002:**
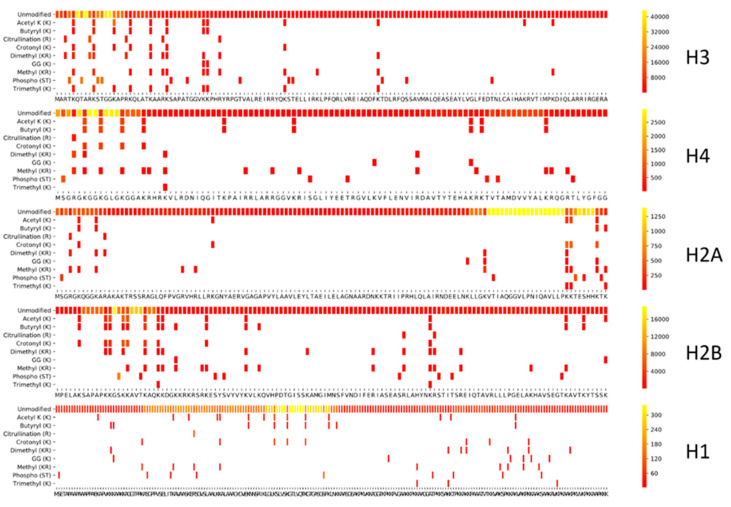
Theoretical coverage of the histone code. A spectral library containing all histone peptidoforms with up to 3 tryptic missed cleavages was predicted using MS²PIP. Next, all delta masses of curated histone modifications from uniprot were parsed into this spectral library at their corresponding location, thus creating a searchable MGF file. This was searched using Mascot and the resulting CSV output, including peptide start and end positions, was used to create a histone coverage plot. This represents what is theoretically possible with a probabilistic scoring algorithm, given the set of variable modifications depicted in the Y-axes. The scale to the right of each plot indicates the spectral count (relative to each histone from red to yellow) of each modification/amino acid residue. For predicted spectra, this indicates how many peptidoforms can theoretically be formed with all the known modifications from Uniprot. The 10 amino acid marker functionality was disabled in the figures produced in this manuscript. The histone variants depicted are H3.2, H2A1, H2B1, and H1.2.

**Figure 3 proteomes-09-00017-f003:**
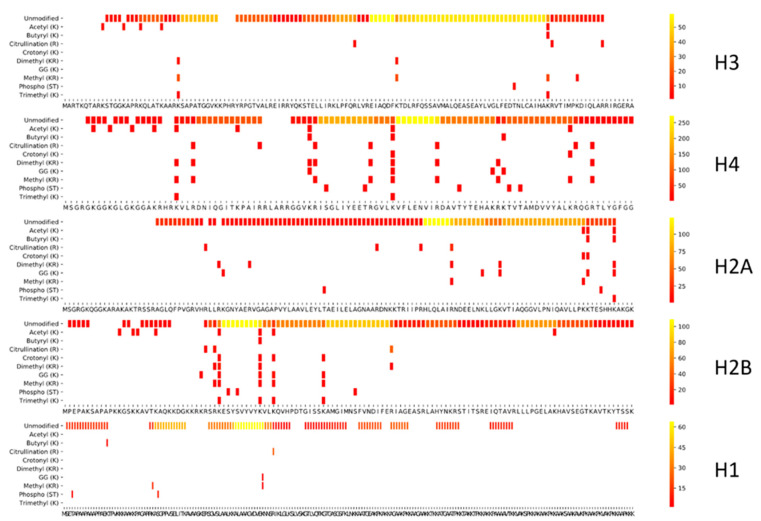
Histone coverage of a tryptic digestion. To assess the coverage of an actual experiment, five replicate tryptic digests were merged and searched, allowing four missed cleavages and the nine PTMs depicted in the theoretical search from Figure 2. The histone variants depicted are H3.2, H2A1, H2B1 and H1.2.

**Figure 4 proteomes-09-00017-f004:**
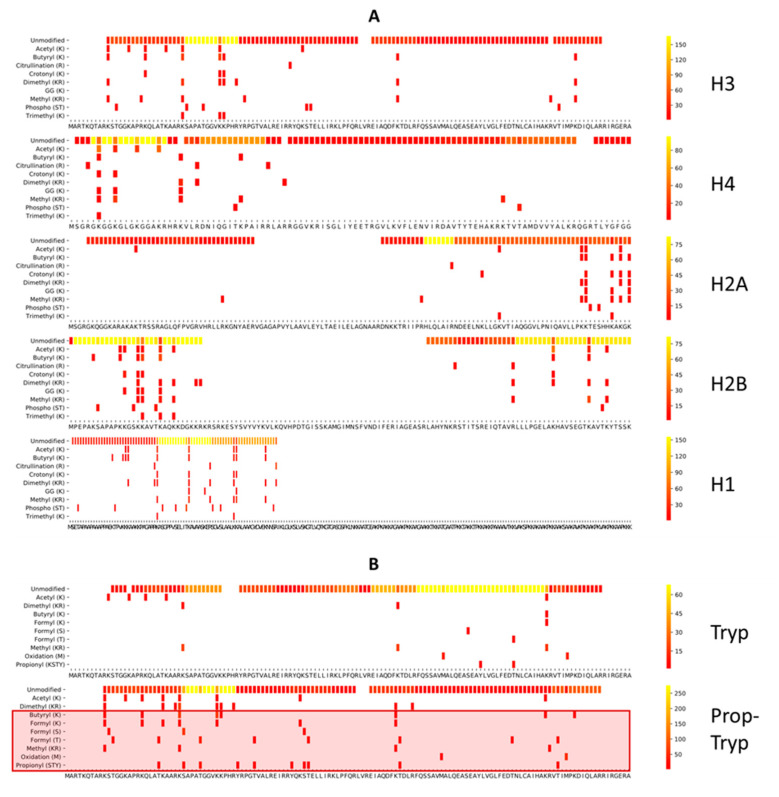
Coverage plot of a tryptic digest following propionylation. (**A**) When histone lysines and N-termini are derivatized with, e.g., propionic anhydride, the lysine cleavage sites are blocked with propionyl groups and the coverage of the histone code shifts considerably, with much-improved sampling of the H3 and H4 N-tails. The histone variants depicted are H3.2, H2A1, H2B1 and H1.2. (**B**) Visualization of chemical noise introduced on histone H3.2 through derivatization. Several non-biological modifications are introduced during sample preparation. The red box highlights different PTMs that are (potentially) changed by the PropTryp workflow. These can no longer be interpreted in a biological context and increase ambiguity and false discovery rate.

**Figure 5 proteomes-09-00017-f005:**
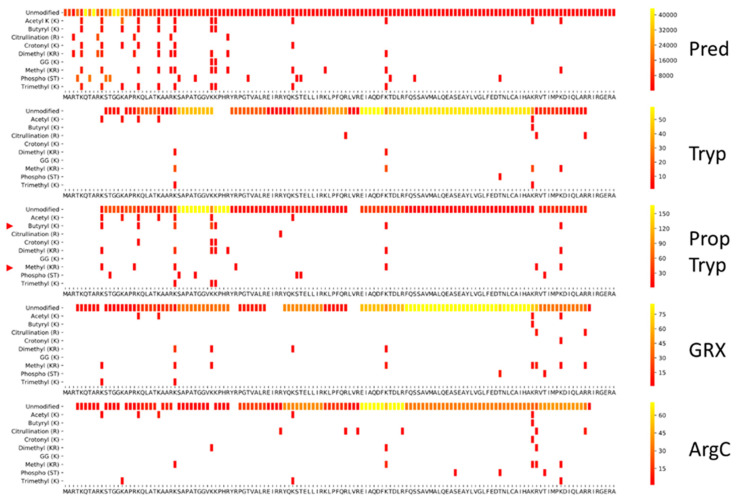
Comparison of the histone H3 PTM coverage for the different workflows. From top to bottom: Pred: theoretically predicted tryptic spectral library of H3.2 searched with the PTMs depicted in the Y-axis; Tryp: empirical digest with trypsin; PropTryp: tryptic digest following derivatization with propionic anhydride (arrowheads indicate PTMs impacted by the derivatization); GRX: empirical digest with GingesRex enzyme; ArgC: empirical digest with ArgC enzyme. The latter three were all searched with ArgC as enzyme specificity. Coverage plots of all other histones are presented in Appendix A.

**Figure 6 proteomes-09-00017-f006:**
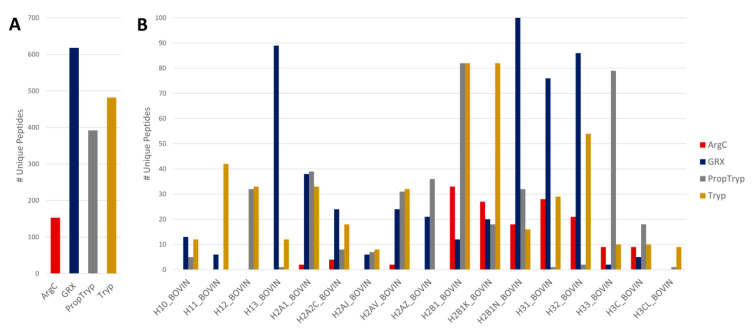
Number of unique histone variant Peptide spectral matches (PSMs) identified by each protocol. (**A**) Unique PSMs that can only be attributed to one histone variant are flagged by Mascot as Unique. Overall, GRX yields the most unique peptides over all variants. (**B**) Unique peptides split up into numbers per histone variant identified. The absence of unique peptides implies that the given protocol cannot uniquely identify this variant. Note that densely sampled stretches with different peptide charge states and modifications appear emphasized in this representation. This is clearly demonstrated by, e.g., H31 and H33 variants called most efficiently in GRX by different charge states and modifications of H3_84–106_.

**Figure 7 proteomes-09-00017-f007:**
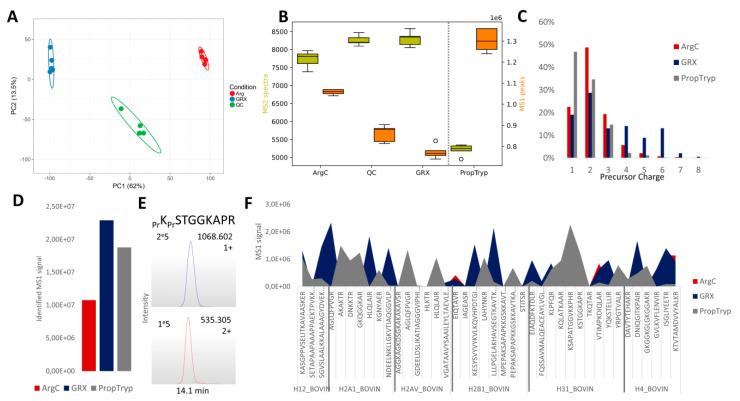
In-depth analysis of three different enzymatic treatments with Arg-C specificity. (**A**) The Principle Component Analysis of the feature abundances from the Progenesis GRX_ArgC project containing ArgC (red) and GRX (blue) shows a clear discrepancy between both workflows despite their identical cleavage specificity. The QC (green) is a mixture of both conditions interspersed throughout the sample list [27]. (**B**) This double-sided boxplot shows the inverse pattern of MS1 peaks (orange, right axis) and MS2 spectra (blue, left axis). PropTryp has the least MS2 spectra, while it has the most MS1 precursors. Grey line indicates the separate Progenesis PropTryp project. (**C**) Charge state distribution of the MS1 precursors from (**B**) for the three workflows, showing a higher proportion of singly charged precursors in PropTryp, which are not selected by the instrument for fragmentation in DDA mode. (**D**) Summed MS1 signal of all identified peptidoforms showing that the highest portion of the generated ions is being annotated in GRX, while all treatments were done on the same starting amount of bovine histones. (**E**) The histone H3K9-R17 peptide stretch XICs show that the singly charged (up) ion species comprises double the amount of ions compared to the doubly charged (down) counterpart. Intensities are depicted in the upper left corner, and retention time is shown in the X-axis. (**F**) MS1 signal intensity for each peptide sequence annotated in the different treatments showing the complementarity between PropTryp and GRX.

**Figure 8 proteomes-09-00017-f008:**
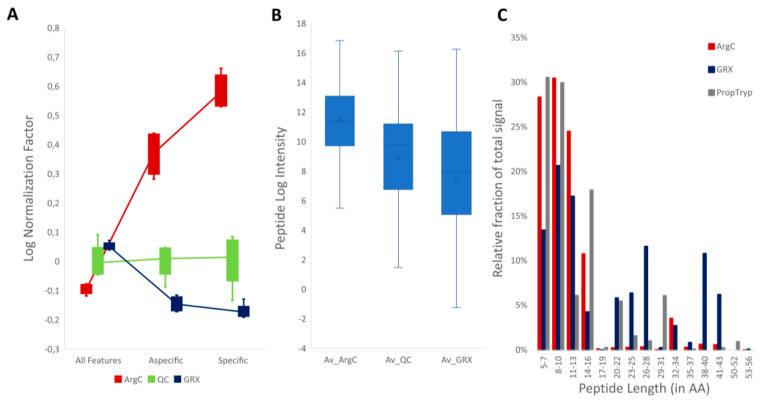
In-depth enzyme specificity and efficiency. (**A**) Calculated logarithmic normalization factors based on all features (left), annotated peptidoforms searched with aspecific cleavage (middle) and annotated peptidoforms searched with specific cleavage (right). The axis depicts normalization factors and inversely depicts the lower signal in ArgC. (**B**) Signal intensity (y-axis) of identified peptidoforms with up to two missed cleavages, which are indicative of reduced enzyme efficiency. (**C**) Distribution of peptide length for all workflows.

**Figure 9 proteomes-09-00017-f009:**
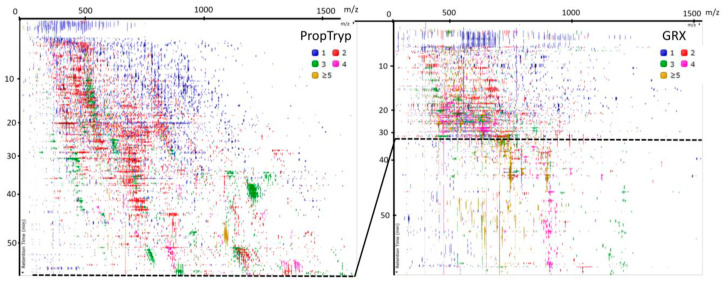
LCMS 2D representation of eluting features of PropTryp and GRX. The Y-axis depicts the retention time from top to bottom as a function of MS1 scan rate (not linear), and the X-axis represents *m*/*z*. Features are color-coded according to their charge state. The dotted line in GRX depicts the point beyond which peptides with propionyl groups become too hydrophobic to be measured with the LC setup applied in this study. This illustrates that a set of longer and higher charged peptides are better detectable with GRX.

**Figure 10 proteomes-09-00017-f010:**
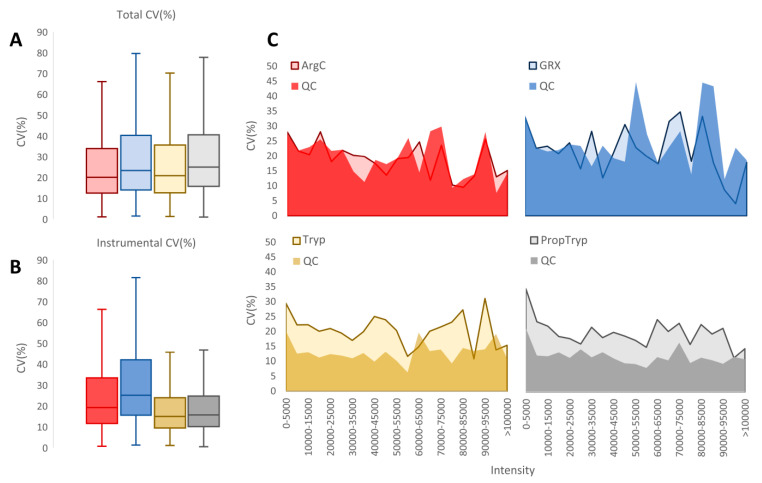
Variability assessment of the workflows. (**A**) Boxplot of the total % covariance (%CV) as calculated from the separately prepared and measured samples of each workflow. (**B**) Boxplot of the instrumental %CV as calculated from the respective quality control (QC) data of each workflow. (**C**) Total %CV (light-colored graph) and instrumental %CV (dark-colored graph) as a function of precursor ion abundance for each workflow.

## Data Availability

The mass spectrometry proteomics data have been deposited into the ProteomeXchange Consortium via the PRIDE partner repository with the dataset identifier PXD024423 and 10.6019/PXD024423.

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
