# Peer review of "Histone Sample Preparation for Bottom-Up Mass Spectrometry: A Roadmap to Informed Decisions"

_proteomes, 2021, doi:10.3390/proteomes9020017_

Round 1
Reviewer 1 Report
As a reviewer, I appreciate the authors’ effort to improve the analysis of PTMs of histone proteins using LC-MS/MS, which is much needed to understand the profiles of epigenetic modification and regulations of histones.
However, histone proteins, core histones, H2A, H2B, H3, and H4, and linker histone H1, consist of a group of high-related subtypes within each group. Especially histone H1 has 7 somatic variants.
The correct identification of peptides belonging to each histone variant can be very challenging for the highly similar regions. I don’t see the authors have specifically addressed this challenge. I would suggest the authors improve their model to identify the PTMs on specific histone variants before it can be accepted for publication.
Reviewer 2 Report
The authors of this manuscript, “Histone sample preparation for bottom-up mass spectrometry: a roadmap to informed decisions”, present a tool for annotating histone tail post-translational modifications (PTMs) and benchmark the detection rate and reproducibility of several histone preparation protocols. The authors sufficiently demonstrate the differences using both conventional and a novel histone preparation protocol, using their new code for generating useful figures for visualizing coverage of histone tails. There is quite a bit of deeply informative, but somewhat nuanced, discussion of the implications of using these different sample preparation approaches that is interesting but difficult to parse. Overall, the authors are clearly knowledgeable of the histone PTM field with good discussion of the current challenges and limitations in preparing and analyzing histone PTMs. I imagine the manuscript’s conclusions will be of interest to mass spectrometrists performing histone PTM work.
I recommend publication of the manuscript with the following major and minor revisions to support the interpretation and the presentation of the manuscript’s main points.
Major issues:
- The authors make many interesting, but not well explained, claims regarding fundamental LC-MS properties of histone peptides. The merit of these ideas is strong, but needs much better clarification within such a dense manuscript. The most major of these points is the recurring discussion of DDA and precursor ions. I’ve summarized some of these confusing parts below:
- Pg 8 lines 290-291 the authors state that “[this] hampers accurate identification and localization of PTM combinations with current search algorithms”. What will hamper the ID and localization? Longer peptide lengths? Why does that make ID and localization harder?
- Much discussion in section 3.2.2 Feature detectability (pg 10, lines 351-358) is very unclear. Is this MS1 fingerprinting? Are the features even peptides?
- If sampling +1 is such an issue with DDA (Figure 6B), why not use the lab’s own hSWATH method, which would fragment any precursor within the SWATH range?
- Pg 13 lines 445-446 advocates for “alternative fragmentation and annotation strategies”, but this section appears to be addressing retention time. Wouldn’t this instead suggest alternative liquid chromatography methods? This suggestion for alternative fragmentation and annotation should be explained further for it to make sense in the context of the paragraph.
- Pg 15 lines 496-498 discusses the charge state distribution produced by propionylation being not amenable to DDA -- this seems to suggest using charge state-independent sampling, like the authors’ hSWATH method, rather than the poor suitability of propionylation for histone PTM work.
- Figure 1 requires clarification either in the text or in visual design as it’s unclear how to interpret this figure. Specifically, it would benefit from a schematic workflow depiction, rather than the current “nodes and edges” representation. Commonly this type of figure shows something like an eppendorf at the “beginning”, then the three four sample preparation approaches, then acquisition (perhaps including some representation for what the QCs are at this point) and finally the analysis.
Minor points:
- Pg 2 line 49 uses unusual abbreviations for retention time and drift time, which are more commonly RT and DT respectively.
- Pg 4 line 163 states that “there was no need to predict retention times” -- is the MS2PIP algorithm capable of predicting retention times for multiply-modified peptides? There should be some text to clarify if this is possible but was not necessary, or if it is not possible at all.
- Pg 4 line 164 GitHub link is not set to “public”. Please open up the repository and provide the proper usage license.
- Pg 5 line 221-223 describes an in silico predicted library for multiply-modified histone peptides. From my understanding, this library was not actually predicted completely, but rather the unmodified peptide sequences were predicted and then the m/z shifts from modifications were added into the “base” peptide sequence spectra. If that is the case, the authors should explain this a bit clearer, as the text currently implies that MS2PIP predicted the multiply-modified histone peptides completely.
- Figures 2-5 would benefit from including the residue number on the x-axis, perhaps below the amino acid letters.
- Figure 2. It’s unclear to me how you got spectral counts from an in silico library. Please add some text explaining how this was done.
- Figure 6D, E, and F y-axes should be labeled.
- Figure 7A, B C should label y-axes, Figure7C should label x-axis
- Figure 7B is very confusing, perhaps a better representation would be a box-and-whisker rather than the line graph.
- Figure 9 requires axis labels.
- Acknowledgement section seems like it could be removed, as it’s just template text.
Reviewer 3 Report
This work dealt with LC-MS/MS analyses of peptides formed by cleavage of a model histone mixture from bovine thymus. The aim was to compare protocols based on the application of different proteolytic approaches to analyze post-translational histone modifications such as acetylation, methylation and phosphorylation. In total, there were four experimental procedures developed utilizing the following proteases: trypsin (without any chemical modification of the proteins or with a modification of Lys residues by propionylation), Arg-C and GRX. Samples were run in several replicates and pooled to suppress system variability, also mixed controls were prepared. The determined modification patterns for H3, H4, H2A, H2B and H1 histones were compared with their theoretical counterparts constructed using available modification data from the UniProt database. It has been shown that trypsin and GRX are suitable for histone analysis because of their specificity and efficiency. Trypsin application combined with lysine propionylation is a good choice to cover the N-terminal modification of H3 and H4 despite of a disturbing chemical noise, which is formed concomitantly, and a shift in charge-state distribution of peptides decreasing the number of MS/MS precursors. Arg-C was found less specific, which hampered accurate quantification. As a complementary enzyme, GR showed an advantage of providing peptides of a larger size.
This manuscript is well written and supported by relevant data. Interpretation as well as discussion of the results is adequate. Conclusions are well drawn.
I would have only a few minor points to be addressed:
Line 113, please specify the composition of the activation solution
Line 273, by comparing Figs. 2 and 3, I see the most striking difference for histone H1, where only a minimum of experimentally detected PTMs is shown in Fig. 3; the authors should provide a comment to the results with H1.
Lines 308-309; please explain how it is possible to convert a methylated lysine to butyryl-lysine by propionylation; I think this is a misinterpretation of the mass change, which was registered
Reviewer 4 Report
The manuscript submitted by Daled et al. is interesting and useful for those who want to study histone post-translational modifications using bottom-up strategies. It compares different methodologies analyzing the points in favor and against in each case. It also provides data on the use of a new proteolytic enzyme that specifically cuts at Arg C-terminus. The manuscript is detailed and well written. However, several points need to be improved before it is suitable for its publication at Proteomes.
A major aspect that needs the author's attention is the coverage of individual histone H1 subtypes, which must be addressed in the text and included in the figures. According to the data in the supplementary Table 1, in the Sigma histone preparation four histone H1 subtypes, H1.0, H1,1, H1.2, and H1.3. Some of the peptides may be common to several subtypes, while others are unique.
Minor points:
- The quality of Figures 2-5 must be improved, letters are out of focus and too small to read.
- The abbreviation used in line 94 is not defined.
- Citation 15 does not belong to Lee et al.
- The term aspecific is not used correctly in the text.
Round 2
Reviewer 4 Report
The revised manuscript is significantly improved, as this version addresses the analysis of histone subtypes/variants. However, there are minor details that must be corrected before its publication:
- The correct annotation of H1 subtypes is H1.0, H1.1, H1.2, and H1.3.
- The specific subtype or variant whose sequence appears in figures 2, 3, and 4, as well as in the supplementary figures must be annotated in the captions.
- In the caption of Figure 3, instead of Figure 1, it should be Figure 2.
